# Genetic similarity between relatives provides evidence on the presence and history of assortative mating

Hans Fredrik Sunde [1,2] ✉, Nikolai Haahjem Eftedal [3], Rosa Cheesman[3], Elizabeth C. Corfield [4,5], Thomas H. Kleppesto [1,6], Anne Caroline Seierstad[3], Eivind Ystrom [3,5], Espen Moen Eilertsen [3] & Fartein Ask Torvik [1,3]

Assortative mating – the non-random mating of individuals with similar traits – is known to increase trait-specific genetic variance and genetic similarity between relatives. However, empirical evidence is limited for many traits, and the implications hinge on whether assortative mating has started recently or many generations ago. Here we show theoretically and empirically that genetic similarity between relatives can provide evidence on the presence and history of assortative mating. First, we employed path analysis to understand how assortative mating affects genetic similarity between family members across generations, finding that similarity between distant relatives is more affected than close relatives. Next, we correlated polygenic indices of 47,135 co-parents from the Norwegian Mother, Father, and Child Cohort Study (MoBa) and found genetic evidence of assortative mating in nine out of sixteen examined traits. The same traits showed elevated similarity between relatives, especially distant relatives. Six of the nine traits, including educational attainment, showed greater genetic variance among offspring, which is inconsistent with stable assortative mating over many generations. These results suggest an ongoing increase in familial similarity for these traits. The implications of this research extend to genetic methodology and the understanding of social and economic disparities.

Assortative mating – the non-random pairing of individuals with similar traits – has long been a challenging topic of interest across various fields, including genetics[1–9], sociology[10–12], and economics[13,14]. Consequences of assortative mating are wide-ranging, affecting topics such as genetic research methods[15,16], relationship quality[10,17,18], and the perpetuation of social and economic inequalities[10,13,14]. Although partner similarity have been documented for numerous characteristics[15,16,19], it remains uncertain to what extent these similarities result from assortative mating or other processes, such as convergence over time[18,20].

Hence, the genetic consequences are unknown. Recent advances in data availability have enabled empirical investigation into the genetic consequences of assortative mating, wherein two are of key interest: First, partners should exhibit genetic similarity for assorted traits; and second, genetic similarity between relatives should increase for the assorted traits in subsequent generations[1–3]. In this paper, we aim to: 1) clarify the theoretical consequences of assortative mating on genetic similarity in extended families; 2) use polygenic indices to assess trait-specific genetic similarity between partners for a range of psychosocial,

[1]Centre for Fertility and Health, Norwegian Institute of Public Health, Oslo, Norway. [2]Department of Psychology, University of Oslo, Oslo, Norway. [3]PROMENTA Research Center, Department of Psychology, University of Oslo, Oslo, Norway. [4]Nic Waals Institute, Lovisenberg Diakonale Hospital, Oslo, Norway. [5]PsychGen Centre for Genetic Epidemiology and Mental Health, Norwegian Institute of Public Health, Oslo, Norway. [6]Department of Psychology, Norwegian University of Science and Technology, Trondheim, Norway. ✉e-mail: hansfredrik.sunde@fhi.no

anthropometric, and health-related traits; 3) investigate whether these traits also exhibit increased genetic similarity among relatives; and 4) use the observed genetic similarity in mother-father-child trios to investigate the stability of assortative mating over many generations.

According to a recent meta-analysis, phenotypic correlations between partners exist for many traits[16]. The correlations are particularly high for cognitive and social traits like educational attainment (0.53) and political values (0.58), but moderate correlations exist for many diverse traits such as height (0.23), depression (0.14), and personality (0.08–0.21). Positive correlations between partners can arise from numerous processes, including convergence (partners becoming more alike over time due to mutual influence), common environments (partners originating from similar environments that affect their traits, but without influencing partner formation), and assortative mating (individuals tending to form partnerships with those having similar traits)[18]. If partner similarity arises because of assortative mating, then this will induce cross-partner correlations between factors that are associated with the trait. If the trait is heritable – which most traits are[21,22] – then partners will tend to carry genetic variants with similar effects on the trait. Genetic similarity between partners has been documented for some traits, including height and educational attainment[6,19,23–25]. For example, Yengo et al.[25] investigated genetic similarity in partners from the UK Biobank across 32 complex traits, but lack of statistical power left the question unresolved for most traits. Here, we remedy this by investigating partners in the Norwegian Mother, Father, and Child Cohort Study (MoBa)[26,27], the largest cohort of confirmed partners with available genetic data (n = 47,135).

If assortative mating leads to genetic similarity between partners, then any resulting offspring are likely to inherit trait-specific genetic variants with similar effects from both parents. This has two important consequences: First, the trait-specific genetic variance in the population will increase because genetic variants with similar effects will tend to co-occur in the same individuals (i.e., variants will be in linkage disequilibrium)[3,8,28,29]. Second, trait-specific genetic similarity between relatives will increase because other family members are more likely to inherit genetic variants with similar effects[1–4,8]. With no assortative mating, genotypic correlations between family members for a trait should equal the coefficient of relationship. For example, full siblings (not including monozygotic twins) and parent-offspring pairs are first-degree relatives, with a coefficient of 0.50; aunt/uncle-niece/nephew and grandparent-grandchild pairs are second-degree relatives, with a coefficient of 0.25; and first cousins are third-degree relatives, with a coefficient of 0.125. Under assortative mating, however, the trait-specific genotypic correlations will be higher than the corresponding coefficients of relationship. Importantly, assortative mating only induces correlations between trait-associated loci and should not be confused with inbreeding, which induces correlations between all loci[30]. With successive generations of stable assortative mating, trait-specific genetic variance and genotypic correlations between relatives increase asymptotically towards an equilibrium, at which point they become constant across generations[3,8,28,29]. (See also Supplementary Note 2).

In this paper, we study the extent of assortative mating on a range of phenotypes and its historical consequences by using genetic data from extended family members. Our first aim is to derive the expected genotypic correlations between family members under various assumptions using path analysis. There are earlier theoretical papers that lays out the consequences of assortative mating on familial resemblance[1–5,31]. However: 1) they often focus on phenotypic rather than genotypic resemblance; 2) they don't consider imperfectly measured genetic factors (i.e., polygenic indices); 3) they often do not consider gene-environment correlations; and 4) they either don't consider disequilibrium or do so only under simplistic assumptions. We use path analysis because it offers a ready way to relax assumptions while making the theory accessible for non-specialists. In doing so, we describe a general formula for finding such correlations between any two extended family members under assortative mating at equilibrium. Our results imply that genetic similarity between distant relatives should be more affected by assortative mating than similarity between close relatives[1,3,32]. Our second aim is to document polygenic index correlations for various traits among partners in MoBa[26,27]. We find genetic evidence of assortative mating for nine out of sixteen investigated traits. Our third aim is to investigate whether genetic similarity between

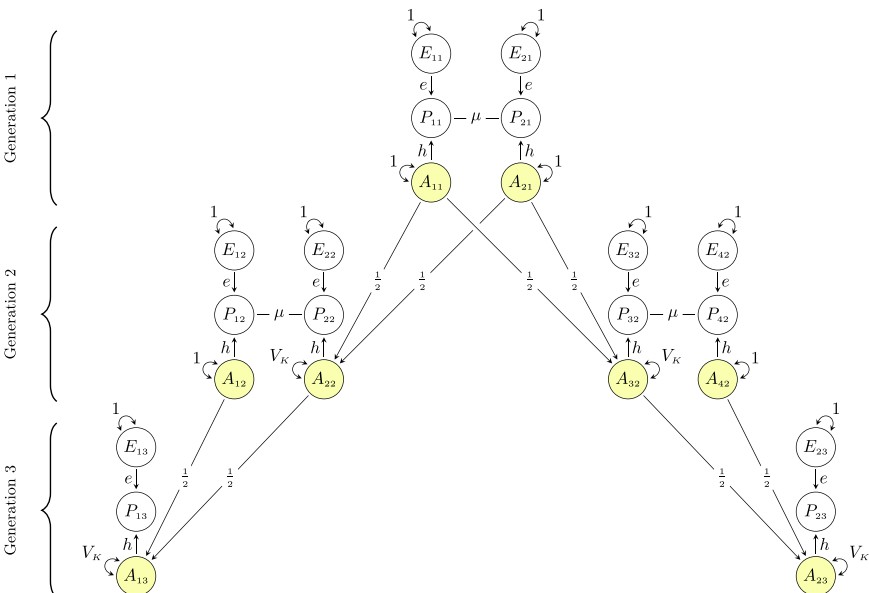

**Fig. 1 | Path diagram of similarity in extended families under assortative mating.** Path diagram for a model of genetic similarity in extended families under phenotypic assortative mating at intergenerational equilibrium (i.e., equal variance across generations). The partner correlation attributable to assortment is denoted by $\mu$, the recombination variance is denoted by $V_K$, and $h$ and $e$ denote the effect of additive genetic ($A_{it}$) and environmental factors ($E_{it}$), respectively, on the phenotype ($P_{it}$) of individual $i$ in generation $t$. All variables have unit variance, meaning $e = \sqrt{1 - h^2}$ and $V_K = \frac{1 - \mu t h^2}{2}$. See the Supplementary Notes 1–3 for path diagrams with relaxed assumptions.

relatives was increased as predicted for these traits. We find that polygenic index correlations among relatives was increased in a way that broadly corresponded to the theoretical expectations. Trait-specific genetic similarity between partners and elevated genetic similarity between relatives indicate that many of the previously observed phenotypic correlations are partly attributable to assortative mating. Our fourth aim is to use mother-father-child trios to test whether the observations were consistent with equilibrium. Although some traits did not significantly deviate from equilibrium expectations, psychosocial traits like education attainment did. This would imply that that the genetic variance and genetic similarity between relatives for these traits are still increasing across generations.

## Results

Figure 1 shows a theoretical model of similarity in extended families in the presence of assortative mating at intergenerational equilibrium. The model includes eight individuals ($i$) in three generations ($t$): two partners in the first generation, their two children in the second generation (who are each other's full sibling) along with their respective partners, and two children in the third generation (who are each other's first cousin). The phenotype that is assorted on is denoted with $P_{it}$, whereas trait-associated additive genetic factors and unique environmental factors are denoted with $A_{it}$ and $E_{it}$, respectively. The genotypic correlation between any two individuals is the sum of all valid chains of paths between their respective additive genetic factors and the value of a single chain is the product of its path coefficients[33,34]. Valid chains always begin by tracing backward ($\leftarrow$) in relation to the direction of arrows, incorporating exactly one double-headed arrow ($\leftrightarrow$), after which tracing continues in a forward direction ($\rightarrow$). Because the variables in Fig. 1 have unit variances, all valid chains connecting a variable to itself will sum to 1, allowing us to immediately trace in a forward direction (i.e., change direction at once). Copaths ($-$), which are arrowless paths representing associations arising from assortment[35], link together valid chains per the rules above, forming longer, valid chains. For a more detailed description of path tracing rules involving copaths, see Balbona et al.[36] or Keller et al.[37] Path diagrams with relaxed assumptions (e.g., gene-environment correlations) are presented and discussed in Supplementary Notes 1–3, whereas simulations validating our theoretical expectations are presented in Supplementary Notes 4 and 5.

### Expected genotypic correlations in the nuclear family

In Fig. 1, there is only one valid chain between partners' additive genetic factors (e.g., $A_{11} \leftrightarrow A_{21}$): $h \times \mu \times h$. The genotypic correlation between partners (denoted $\rho_g$) is thus the phenotypic correlation attributable to assortative mating, $\mu$, weighted by the trait's heritability, $h^2$:

$$\rho_g = \mu h^2 \tag{1}$$

Similarly, we can trace the valid chains between the additive genetic factors of a parent and their offspring (e.g., $A_{11} \leftrightarrow A_{22}$). There are two valid chains: one directly from parental genetic factors to offspring genetic factors, $\frac{1}{2}$, and one through the other parent via the assorted phenotype: $h \times \mu \times h \times \frac{1}{2}$. The genotypic correlation between parent and offspring is therefore $\frac{1}{2} + \frac{h\mu h}{2}$. With no assortative mating ($\mu = 0$), this reduces to $\frac{1}{2}$. For siblings ($A_{22} \leftrightarrow A_{32}$), there are four valid chains: $\frac{1}{4} + \frac{1}{4} + \frac{h\mu h}{4} + \frac{h\mu h}{4}$, which can be rearranged so that it equals the genotypic parent-offspring correlation. Because they are equal, we can define a common denotation ($r_{g_1}$) for first-degree relatives. We can also substitute $h \times \mu \times h$ with $\rho_g$ giving us:

$$r_{g_1} = \frac{1 + \rho_g}{2} \tag{2}$$

In other words, the genotypic correlation between first-degree relatives, $r_{g_1}$, is increased by half the genotypic correlation between partners at equilibrium. (Note that the phenotypic correlation will not be the same for siblings and parent-offspring despite the same genotypic correlation[3]). An advantage of using path analysis is how easy path diagrams are to expand. In the Supplementary Information, we detail how relaxing the assumption of equilibrium (Note 2) and including polygenic indices (Note 3) changes the correlations. During disequilibrium, the genotypic correlation between partners will still conform to Eq. (1), but the correlation between relatives will be less than what Eq. (2) would predict. For polygenic index correlations, one must include a term representing the imperfect correlation between the polygenic index and the true genetic factor. The polygenic index correlation between partners should therefore be:

$$\rho_{pgi} = \mu h^2 s^2 \tag{3}$$

where $s^2$ is the shared variance between the polygenic index and the true additive genetic factor (i.e., the genetic signal[19]). Assortative mating will induce covariance between different loci (i.e., linkage disequilibrium), which is included in the genetic signal. This means that $s$ may be larger than the correlation between the true direct effects and the polygenic index weights, and as such do not represent the accuracy of the polygenic index weights (see Supplementary Notes 3, 4.4, and 5.6). If the genetic signal is low, the polygenic index correlation between partners will be biased towards zero compared to the true genotypic correlation[19]. For first-degree relatives, the equation becomes similarly altered, but because the error terms in the polygenic indices are correlated between relatives, the polygenic index correlation will be biased towards the coefficient of relatedness rather than zero:

$$r_{pgi_1} = \frac{1 + \rho_{pgi}}{2} \tag{4}$$

### Expected genotypic correlations in the extended family

The model in Fig. 1 has two properties that allow a general algorithm to find the expected genotypic correlation between any two members in extended families. First, all the chains that connect the genotypes of first-degree relatives can readily be continued without breaking path tracing rules. Second, all chains between the genotypes of any two related individuals are mediated sequentially through the genotypes of first-degree relatives. The genotypic correlation between $k^{th}$-degree relatives, denoted $r_{g_k}$, can thus be attained by raising the genotypic correlation between first-degree relatives to the degree of relatedness:

$$r_{g_k} = \left(\frac{1 + \rho_g}{2}\right)^k \tag{5}$$

For example, the expected genotypic correlation between third-degree relatives like first cousins is $(\frac{1+\rho_g}{2})^3$, which can be verified by manually tracing all valid chains between $A_{13}$ and $A_{23}$ in Fig. 1. The genotypic correlation between non-blood relatives like in-laws, which will be non-zero under assortative mating, can be attained by linking together chains of $r_{g_k}$ and $\rho_g$ (for example, $Corr(A_{12}, A_{42}) = r_{g_1}\rho_g^2$). As for polygenic index correlations, they can be approximated by replacing $\rho_g$ with $\rho_{pgi}$ in Eq. (5), although depending on the genetic signal, the true correlation between polygenic indices may be slightly higher (see Supplementary Note 3).

Figure 2 shows how assortative mating changes genotypic correlations between relatives at equilibrium. In Panels A and B, it is evident that assortative mating has a much larger effect on first cousins than full

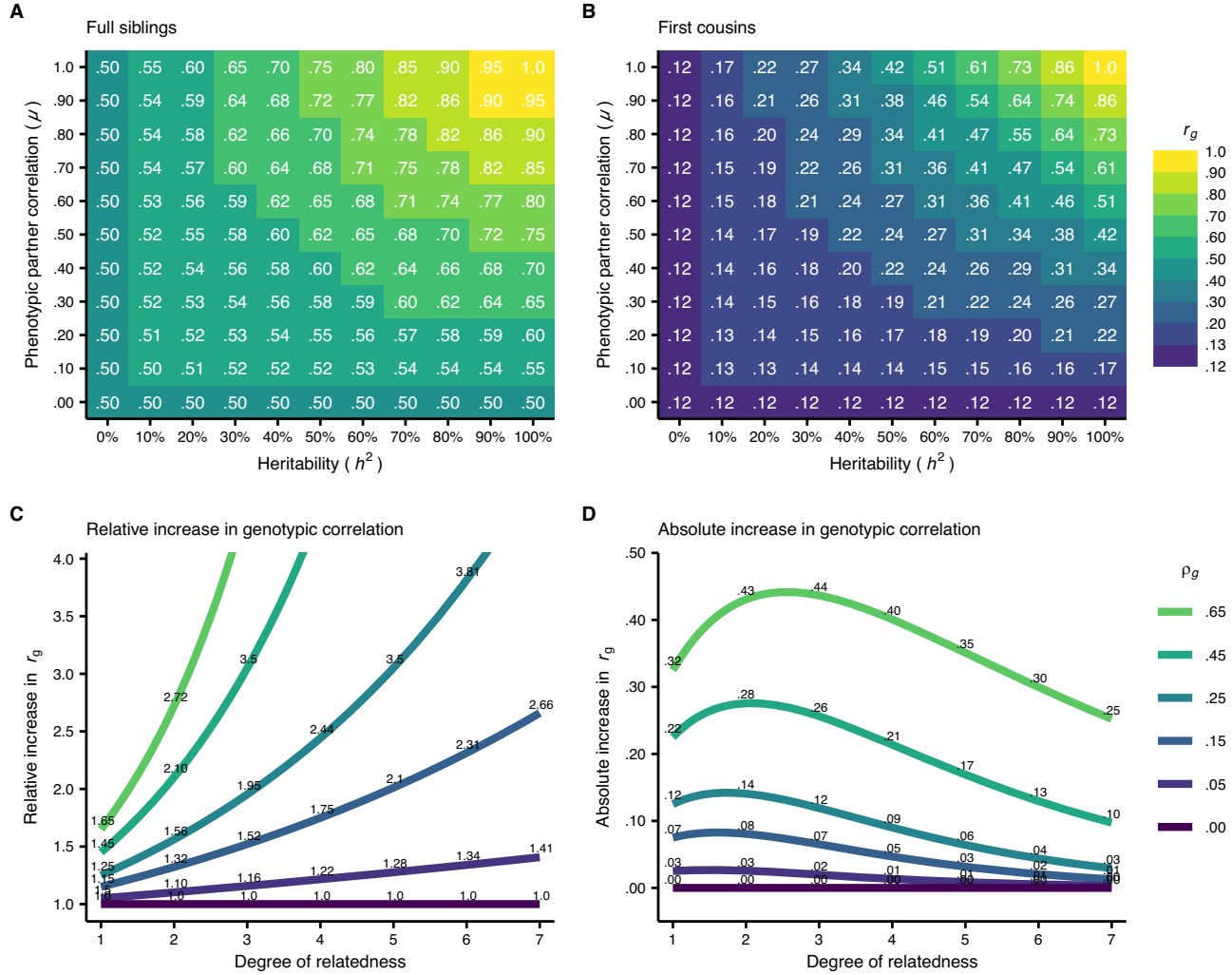

**Fig. 2 | Assortative mating's effect on genotypic correlations between various relatives. A**, **B** The expected genotypic correlation ($r_g$) at equilibrium between full siblings (i.e., first-degree relatives) and first cousins (i.e., third-degree relatives) under different combinations of assortment strengths ($\mu$) and heritabilities ($h^2$). **C**, **D** The relative and absolute increase in genotypic correlation at equilibrium for various relatives and genotypic correlations between partners ($p_g$).

siblings. For example, for a trait where $\mu = .50$ and $h^2 = 50\%$ (meaning $\rho_g = .25$), siblings (Panel A) will have a correlation of $r_{g_1} = .625$ whereas cousins (Panel B) will have a correlation of $r_{g_3} = .244$, reflecting increases of 25% and 95%, respectively, compared to random mating. Panel C shows how this pattern extends to more distant relatives, with the genotypic correlation between second cousins 3.5 times higher than normal if $\rho_g = .25$ ($r_{g_5} = .095$ *vs.* .031). The larger relative increase is not merely because the correlations are smaller to begin with: Panel D shows that the largest absolute increase typically occurs in second-degree relatives like uncles/aunts and nephews/nieces.

The relatively greater increase in correlation between cousins is because third-degree relatives are affected by three assortment processes: Mother-father, uncle-aunt, and grandfather-grandmother partnerships are all correlated under assortative mating and contribute to the increased correlation (Fig. 1). For each additional degree of relatedness, there is an additional assortment process opening pathways for relatives to correlate. This pattern extends to unrelated individuals like siblings-in-laws, who would have a genotypic correlation of $\rho_g r_{g_1} = .157$ if $\rho_g = .25$. It is evident that assortative mating has a relatively larger impact on the genotypic correlation between distant relatives compared to close relatives, and that heritable traits subject to strong assortment can produce significant genotypic correlations between family members who would otherwise be virtually uncorrelated.

## Gene-environment correlations, shared environment, and dominance effects

One limitation with most earlier work, such as Fisher[1], is that they assume a simplistic model where genetic similarity is the only cause of familial resemblance. In Supplementary Note 1, we detail how genetic similarity between relatives are affected by dominance effects, shared environmental effects, and various forms of environmental transmission. If genetic and environmental transmission occur simultaneously, assortative mating will induce (and greatly increase) correlations between genetic and environmental factors. Such gene-environment correlations will, in this context, mimic higher heritability, leading to higher genotypic correlations between partners and thereby exacerbated genetic consequences of assortative mating. However, the relationship between the genotypic correlation between partners and the genotypic correlation between first-degree relatives will stay the same, meaning Eq. (2) and Eq. (4) can be used without making assumptions about gene-environment correlations or other sources of familial resemblance.

This is not the case for distant relatives. If there are substantial shared environmental effects, gene-environment correlations, or other sources of familial resemblance, the properties of Fig. 1 that allow the general algorithm in Eq. (5) are no longer present. This is because non-genetic causes of familial resemblance result in pathways between distant relatives that bypass the genotypes of intermediate relatives,

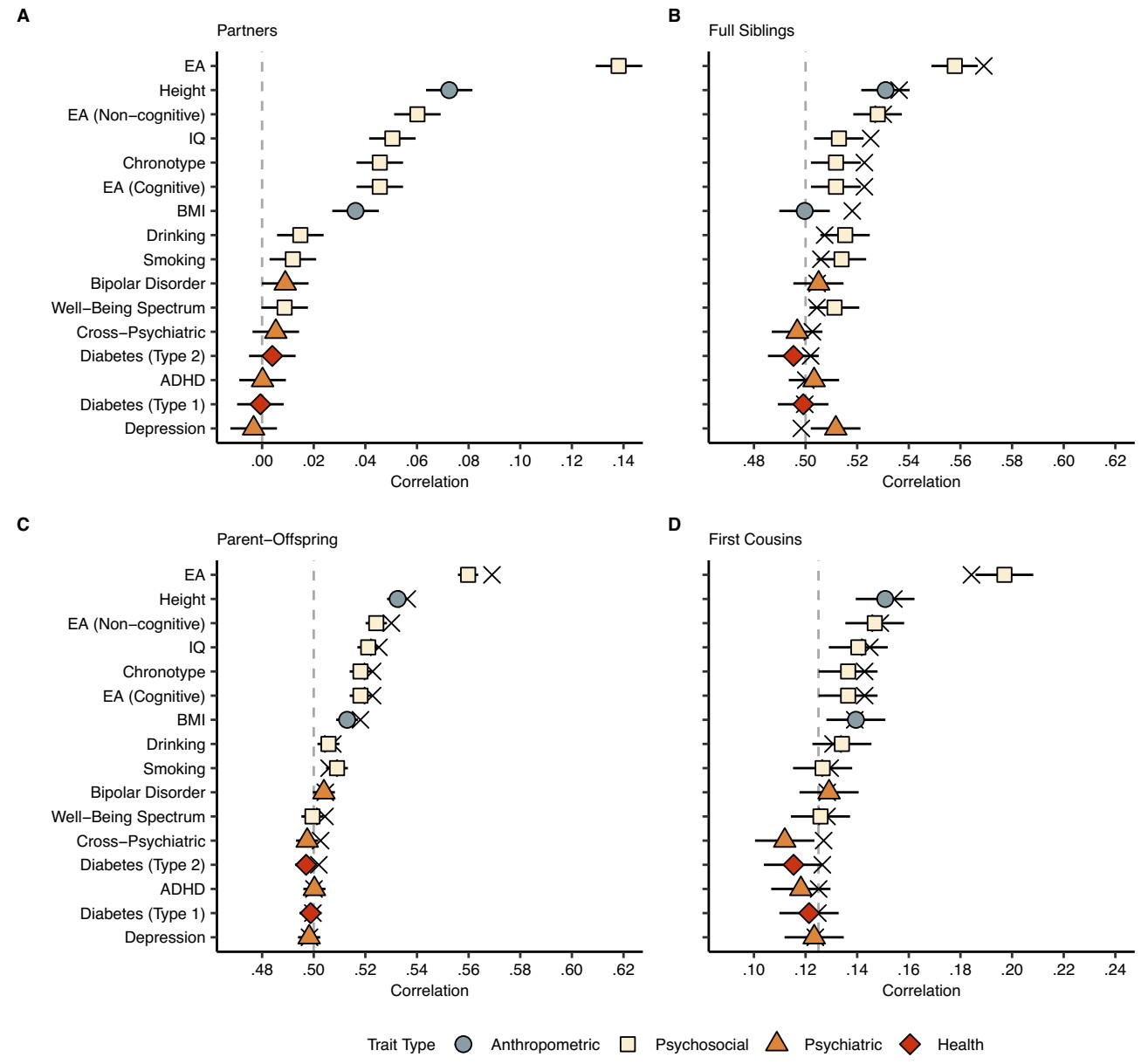

**Fig. 3 | Correlations between family members.** Polygenic index correlations (with 95% CIs) for various traits between various family members: (**A**) partners (*N* = 47,135), (**B**) full siblings (*N* = 22,575), (**C**) parent-offspring (*N* = 117,041), and (**D**) first cousins (*N* = 28,330). The vertical dashed lines are the expected correlation under random mating (i.e., the coefficient of relatedness), and the black crosses are the expected correlation at equilibrium given Eq. (5). Abbreviations: EA educational attainment, BMI body mass index, IQ intelligence, ADHD attention-deficit hyperactivity disorder. Correlations are also reported in Supplementary Table 17.

thus increasing the true genotypic correlation to beyond what Eq. (5) would predict. Equation (5) still serves as a rough approximation, although any statistical model that relies on it could be biased if such extra pathways exist.

**Empirical polygenic index correlations between partners and relatives**

Figure 3 shows polygenic index correlations between family members for a range of traits. Nine out of sixteen traits were significantly correlated between partners (Panel A), including height (0.07), body mass index (0.04), intelligence (0.04), and educational attainment (0.14). When educational attainment was split into cognitive and non-cognitive factors (GWAS-by-subtraction[38]), we find roughly equal partner correlations for both components. Psychiatric traits like

ADHD, depression, cross-psychiatric disorder, and bipolar disorder exhibited no significant correlations between partners. Keep in mind that the correlations will be biased downwards to the extent the genetic signal is poor (ref. Equation (3)).

Panels B, C, and D show polygenic index correlations between full siblings, parents and offspring, and first cousins, respectively (see Supplementary Fig. 29 for other family members). The vertical dashed lines are the expected correlations under random mating and the black crosses are the expected correlations at equilibrium given the partner correlation and Eq. (5). All traits with significant correlations between partners had significantly higher parent-offspring correlations than would be expected under random mating, and we observed similar patterns for other relatives. For example, the polygenic index correlation for educational attainment was 0.56

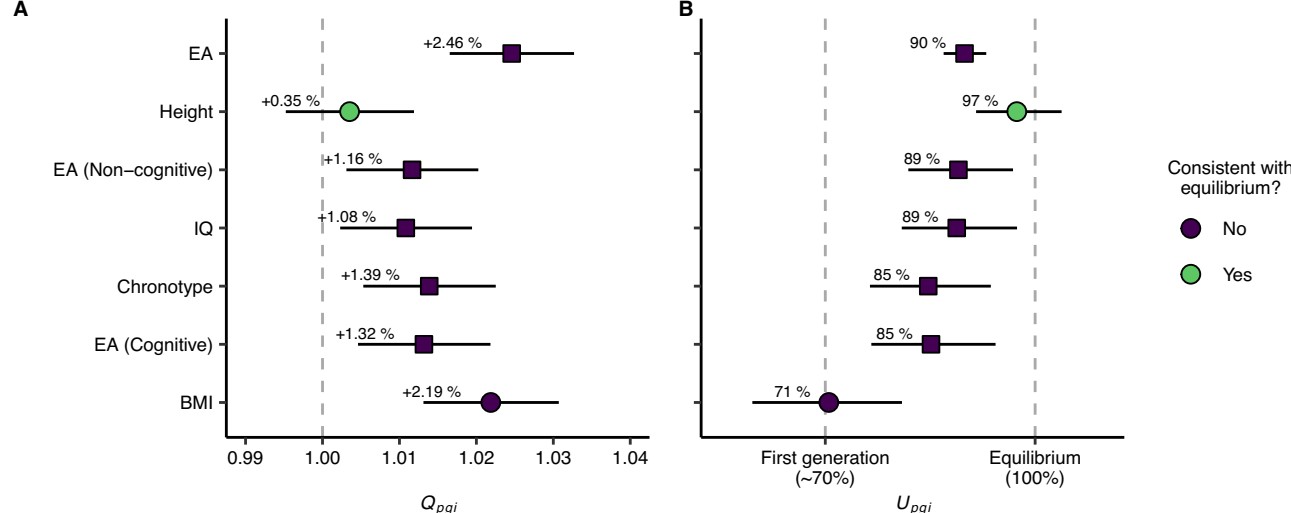

**Fig. 4 | Tests of intergenerational equilibrium.** Parameter estimates (with 95% likelihood-based CIs) from structural equation models using mother-father-child trios (N = 87,896 families, 35,025 of which were complete). **A** Ratio of offspring polygenic index variance to parental polygenic index variance ($Q_{pgi}$, see Supplementary Note 2.1). A value above 1 would indicate that the variance is greater in the offspring generation compared to the parental generation, as expected during disequilibrium. **B** Observed increase in parent-offspring correlation compared to expected increase at equilibrium ($U_{pgi}$, see Supplementary Note 2.3). A value of about 70% would indicate that the parent generation was the first generation to assort on this trait, whereas 100% would indicate that the trait is in intergenerational equilibrium. Only traits with significant correlations between partners are shown. Shape corresponds to trait types in Fig. 3, where circles are anthropometric traits and squares are psychosocial traits. Abbreviations: EA educational attainment, BMI body mass index, IQ intelligence.

(instead of 0.50) between full siblings and 0.20 (instead of 0.125) between first cousins.

## Testing intergenerational equilibrium

We fitted structural equation models using mother-father-child trios to see if a model constrained to equal variance across generations (i.e., equilibrium) resulted in significantly worse fit (see Supplementary Note 6). Six out of nine traits were significantly different from equilibrium. We also investigated two consequences of disequilibrium, namely greater variance in the offspring generation (Fig. 4A) and smaller-than-expected parent-offspring correlations (Fig. 4B). During disequilibrium, the ratio of offspring polygenic index variance to parental polygenic index variance should be positive: $Q_{pgi} = \frac{Offspring\ Variance}{Parental\ Variance}$ (see Supplementary Notes 2.1 and 3.3). However, this ratio is quite sensitive to the genetic signal of the polygenic index, and therefore provides limited information about the history of assortative mating beyond demonstrating disequilibrium. An alternative measure that is less sensitive to the genetic signal is the observed increase in polygenic index correlation as a percentage of the expected increase[19]: $U_{pgi} = \frac{Observed\ Increase}{Expected\ Increase}$ (see Supplementary Notes 2.3 and 3.4). This provides a measure of how close the trait is to equilibrium. By comparing $U_{pgi}$ to reference values under various heritabilities and assortment strengths, it is possible to infer the equivalent number of generations of stable assortative mating if starting from a random mating population. If the parental generation was the first generation to mate assortatively, we would expect $U_{pgi} \approx 70\%$, while we would expect $U_{pgi} = 100\%$ if the trait was in equilibrium.

Height did not deviate from equilibrium: There was no significant difference between the parental and offspring variance nor between the observed and expected correlations. The results for drinking and smoking behavior were also consistent with equilibrium, although the observed partner correlation was too small to make this test informative. Body mass index and other psychosocial traits, on the other hand, did deviate from equilibrium: For example, the polygenic index variance for educational attainment was 2.46% greater in the offspring generation compared to the parental generation. The true genetic

variance ratio is likely much larger: For example, if the polygenic index captures one third of the true genetic factor ($s^2 = 1/3$), then the true variance increase would be approximately 7.4% (see Supplementary Note 3.3). The parent-offspring polygenic index correlation was also slightly but significantly lower than expected at equilibrium ($U_{pgi} = 90\%$, 95% CIs: 87–93%). When we compared this to calculations of what the observed increase would have been after successive generations of assortative mating, we found that $U_{pgi} = 90\%$ is equivalent to approximately three generations of stable assortment (see Supplementary Note 2.3). Results were similar for other psychosocial traits, albeit with somewhat shorter implied histories. Body mass index, on the other hand, had a parent-offspring polygenic index correlation that would imply that the parent generation was the first to mate assortatively ($U_{pgi} = 71\%$, 95% CIs: 60–81%). This would also explain why the sibling correlation – many of whom are in the parent generation – was not higher than expected under random mating.

## Discussion

In this study, our goal was to clarify the theoretical consequences of assortative mating on genetic similarity in extended families and assess empirical measures of genetic similarity to provide insights into the presence and history of assortative mating. We first employed path analysis to deduce the expected polygenic index correlations between relatives under assortative mating. We then presented empirical evidence that assortative mating is present for many traits, leading to significantly increased genetic similarity among relatives for those traits. Finally, we showed that – while assortative mating does not appear to be a recent phenomenon for most traits – genetic similarity is still increasing across generations for psychosocial traits. Here, we discuss the implications of our findings.

Our first aim was to clarify the theoretical consequences of assortative mating. One key finding is the stronger impact of assortative mating on genotypic correlations between more distant relatives. Although not a novel discovery – even Fisher mentioned it offhandedly in his seminal paper[1] – this effect has been largely overlooked in the literature (cf[32].). This is despite important implications. A Swedish economics paper reported that nearly one-third of persistence in

inequality across generations – traditionally attributable to parent-offspring relationships – is attributable to the extended family[39]. Assortative mating's effects on similarity in extended families may be key to understanding these issues. Similar logic may also apply to environmentally mediated sources of similarity[40]. We also described how assortative mating can induce and increase gene-environment correlations, which mimic higher heritability and thereby exacerbate the genetic consequences of assortative mating – especially correlations between distant relatives.

The second aim of this study was to investigate which traits show genetic evidence of assortative mating. One key challenge when evaluating the pervasiveness of assortative mating is that phenotypic partner similarity can come about from multiple processes. Genotypic similarity, on the other hand, can more confidently be attributed to assortative mating. Most anthropometric traits and psychosocial traits had significant polygenic index correlations between partners. The largest correlation was for educational attainment (0.14), which adds to the growing list of evidence that variants associated with educational attainment are undergoing assortative mating[6,19,23,25,41].

Psychiatric traits did not show evidence of assortative mating despite pervasive phenotypic partner correlations[15,16]. Similarly, a recent study found no genetic partner similarity on general risk for psychopathology (i.e., the "p-factor")[42]. These findings seemingly contradict Torvik et al.[19], who reported evidence of assortative mating on depression using a smaller subset of the same cohort. However, that paper used a structural equation model requiring both genetic and phenotypic data, and the polygenic index correlations reported in that paper match those we report here. This could indicate that phenotypic partner similarity in mental health is caused by processes other than assortative mating, such as convergence[20] (which was not modelled in Torvik et al.[19]). On the other hand, the results could also be false negatives resulting from low-quality polygenic indices. The depression polygenic index only correlates .11 with the phenotype in the current cohort[19], meaning the expected partner correlation is only about $.11^2 \times .14 = .0017$ under direct assortment. A false negative is therefore highly likely. Reports of smaller but non-zero phenotypic correlations prior to partner formation suggests that both convergence and assortment play an important role[43,44].

As highlighted in Eq. (3), the polygenic index correlation between partners should be the product of the phenotypic correlation attributable to assortative mating ($\mu$), the heritability ($h^2$), and the genetic signal ($s^2$). If the polygenic index fails to adequately measure the relevant genetic factors (meaning $s^2 \approx 0$), for example due to lack of statistical power or other measurement issues[45] in the underlying genome-wide associations study (GWAS), then the polygenic index correlation will be biased towards zero. The highest observed correlations were for educational attainment and height, which are among the traits with the largest sample sizes in the underlying GWAS. A corollary is that the correlations reported here do not quantify the exact degree of assortative mating because it is confounded by the genetic signal of the polygenic index. Complicating inference further is that the genetic signal is itself increased under assortative mating.

Our third aim was to investigate whether relatives were more genetically similar for traits that exhibit evidence of assortative mating. Our findings broadly correspond to theoretical expectations: Traits with significant polygenic index correlations between partners showed increased similarity between relatives, whereas traits with no correlations between partners broadly exhibit patterns as expected under random mating. These empirical patterns demonstrate the theoretical expectations derived earlier, meaning we should expect distant relatives to be highly correlated for traits under strong assortment. The correlations reported here are underestimated by the quality of the polygenic index, meaning the true genotypic

correlations between relatives are likely much larger. Our findings have at least two implications. First, genetic variants associated with traits undergoing assortment, such as educational attainment, cluster in extended families, thus increasing or maintaining societal stratification by families[39] (i.e. between-family variation); and second, genetic studies that unknowingly involve numerous distantly related individuals may be biased if the genotypic correlations between them are not negligible.

For educational attainment, the polygenic index correlations between first-degree relatives are lower than expected (indicating disequilibrium, see below) while correlations between third- and fourth-degree relatives are higher than expected. This is consistent with substantial gene-environment correlations for educational attainment[46–48]. In Supplementary Notes 1 and 5, we showed theoretically and with simulations that correlations between higher-degree relatives (but not first-degree relatives) will be higher than expected given Eq. (5) if such gene-environment correlations are present.

Our fourth aim was to investigate the history of assortative mating. Our findings differed across traits: Height did not deviate from equilibrium expectations, whereas psychosocial traits such as educational attainment did. This was evident in both lower-than-expected parent-offspring polygenic index correlations and greater variance in the offspring generation. Whether or not a trait is in intergenerational equilibrium has important implications for the consequences of assortative mating because it decides whether differences are increasing across generations or merely maintained. We found that polygenic index variance was stable across generations for height (as well as for traits not undergoing assortative mating). However, psychosocial traits have greater variance in the offspring generation, implying that the traits are in disequilibrium and that assortative mating is currently leading to increased genetic differences in these traits. Although the non-genetic consequences may differ, assortative mating may therefore play a key role in explaining recent increases in inequality[10,13].

Despite being in disequilibrium, the evidence does not suggest that the parental generation was the first to assort on educational attainment. Instead, it appears that the trait is quite near equilibrium. This would also explain the discrepancy between our conclusion and that in Torvik et al.[19], who found no significant deviation from equilibrium using an earlier version of data from the same cohort. We primarily used variance differences across generations whereas Torvik et al.[19] compared the predicted and expected correlations between siblings and partners. Considering that the sibling correlation in Fig. 3B is significantly lower than expected given equilibrium, the change in result likely stem from an increase in statistical power, owing to more genotyped individuals available in the current sample, and further aided by the use of parent-offspring dyads instead of sibling dyads. In this paper, we estimate that the evidence for educational attainment corresponds to approximately three generations of stable, univariate assortative mating starting from a random mating population, but the exact history of assortment will be longer if the strength of assortment has varied over time or if the genotype-phenotype correlation increased for other reasons[49].

Many genetic research methods assume random mating, but our findings suggest that such assumptions are unwarranted for many traits. Accounting for assortative mating poses its own challenges, as the genetic consequences and corresponding methods needed depend on whether assortative mating started recently or has reached intergenerational equilibrium. Studies on the genetics of educational attainment especially – or the many traits that correlate with educational attainment[50] – may therefore be biased unless this is properly accounted for. Twin and family studies that account for assortative mating typically assume equilibrium[37]. For example, Clark[51] uses equations that assume equilibrium when he claims that familial correlations in social class in the United Kingdom can be explained by

genetic similarity alone. Conversely, Kong et al.[46], who investigated genetic nurture effects of educational attainment in an Icelandic sample, assumed no assortment prior to their parental generation. Our findings imply that, for some traits, neither of these assumptions are valid. Although the patterns and history of assortment may be different across populations, future research should investigate how the conclusions from Kong et al.[46] and related papers depend on these assumptions[52,53].

Newer genetic methods that can account for disequilibrium are being developed[36,54]. When these methods are impractical, the potential biases induced by different assumptions must be considered on a case-by-case and method-by-method basis. Different methods will be biased in different ways. For example, assortative mating leads to underestimated heritability in classical twin designs[37,55] and over-estimated heritability in molecular designs[56], with the corollary that the missing heritability problem may be larger than previously assumed[57,58]. Overall, researchers must carefully consider what impacts the presence and history of assortative mating would have on their results.

Despite our large sample size, our results are limited by low-quality polygenic indices, which results in lower partner correlations and consequently less power to detect assortative mating. This is amplified in tests of equilibrium, where smaller polygenic index correlations between partners result in less statistical power to detect deviations from equilibrium. Our tests for equilibrium are therefore less conclusive for traits with small polygenic index correlations, such as drinking and smoking behavior. Furthermore, assortative mating can bias GWAS estimates and thereby bias polygenic indices[59]. Although this should not affect our conclusions (see Supplementary Note 5.6), it does make it difficult to precisely quantify the strength of assortative mating on various traits and hence the magnitude of the genetic consequences.

Another concern is that our results may be confounded by population stratification[60], where (1) the trait in question happens to be more common within certain strata (e.g., subcultures or geographical areas), (2) some genetic variants are randomly present at higher frequencies in these strata, and (3) individuals are more likely to mate within these strata. The combination of the first two phenomena would result in a spurious correlation between those genetic variants and the trait, and when coupled with the third phenomenon, similar spurious correlations could emerge between partners. While we controlled for 20 principal components in our analysis, which is the standard method for addressing stratification[61], this approach may not fully account for this phenomenon[62]. However, the evidence we present aligns well with predictions given assortative mating. It is also not obvious how population stratification could explain increased variance in the offspring generation. Consequently, our results should be considered indicative of assortative mating until a more compelling alternative explanation is offered. Future theoretical work should investigate how the consequences of assortative mating and population stratification differ so that they can better be distinguished in future research.

There are several interesting research avenues that could follow from this work. First, there may be some selection bias in the cohort study our results are based on. Future work using population-wide phenotypic data might provide insights into how much this matters. Second, patterns of assortative mating are likely to vary between populations[63,64], meaning that our empirical findings are not universally generalizable. Replicating these results in other populations will therefore be beneficial. Third, the approach we use here is agnostic as to which trait(s) the polygenic indices actually measure, and which phenotype(s) are being assorted upon. Future research may want to investigate what set of phenotypes mediate the polygenic index correlations between partners, as it may not always be attributable to the phenotype that the polygenic index supposedly measures.

Furthermore, we have assumed assortment is unidimensional. Considering ample evidence of partner correlations across different traits[44,59], future studies may want to extend this line of research to multidimensional assortment.

## Methods
### Sample
We used data from the Norwegian Mother, Father and Child Cohort Study (MoBa)[26]. MoBa is a population-based pregnancy cohort study conducted by the Norwegian Institute of Public Health. Participants were recruited from all over Norway from 1999 to 2008. The women consented to participation in 41% of the pregnancies. Blood samples were obtained from both parents during pregnancy and from mothers and children (umbilical cord) at birth[65]. The cohort includes approximately 114,500 children, 95,200 mothers and 75,200 fathers. The current study is based on version 12 of the quality-assured data files released for research in January 2019. The establishment of MoBa and initial data collection was based on a license from the Norwegian Data Protection Agency and approval from The Regional Committees for Medical and Health Research Ethics. The MoBa cohort is currently regulated by the Norwegian Health Registry Act. The current study was approved by The Regional Committees for Medical and Health Research Ethics (2017/2205).

The sample included all individuals who had been genotyped and passed quality control[27]. This included 77,506 mothers (birth year: M = 1974.36, SD = 5.1), 53,274 fathers (birth year: M = 1972.27, SD = 5.6), and 71,525 children (49% female, birth year: M = 2005.31, SD = 1.94). For the correlations, the sample included 47,135 unique mother-father dyads (i.e., partners). As described in Corfield et al.[27] relatedness relationships in MoBa were inferred from genetic data by applying KING programs[66] to a subset of single nucleotide polymorphisms (SNPs) with call rate < 98% and minor allele frequency (MAF) < 5%. KING accurately infers monozygotic twin or duplicate pairs (kinship coefficient > 0.3540), first-degree (parent-offspring, full siblings, dizygotic twin pairs; kinship coefficient range 0.1770–0.3540), second-degree (half siblings, grandparent-offspring, avuncular relationships; kinship coefficient range 0.0884–0.1770), and third-degree (first cousins; kinship coefficient range 0.0442–0.0884) relationships. This method identified 117,041 parent-offspring dyads, 22,575 full sibling dyads, 35,923 second-degree dyads (e.g., uncle-nephew), 28,330 third-degree dyads (e.g., first cousins), 9392 fourth-degree dyads, and 235,209 dyads of unrelated family members (e.g., in-laws, nephews–uncles' spouses, partners, etc.,) where both members of the dyads had been genotyped and passed quality control.

To test equilibrium, we used all available mother-father-child trios from MoBa. We relied on trios to test equilibrium for the following reasons: 1) It allowed estimating the partner correlation and parent-offspring correlations in the same model; 2) it allowed us to include both the mother-offspring and father-offspring dyads simultaneously thus increasing statistical power; 3) it allowed us to estimate variances separately for the two generations; 4) there was no need to distinguish between correlations between relatives in the parent generation and in the offspring generation, as this is inherent in the design; 5) focusing on the nuclear family removes the need to make assumptions about the genetic signal or gene-environment correlations; and finally, 6) the sample in MoBa is inherently selected on parent-offspring dyads whereas the availability of other relatives is coincidental. Using other relatives, such as siblings, could therefore lead to stronger ascertainment bias. After randomly selecting one offspring from each nuclear family, we were able to construct a sample of 76,869 genotyped mothers, 51,549 genotyped fathers, and 66,751 genotyped offspring, resulting in a total of 87,896 incomplete and complete trios. Of these, 35,025 were complete trios, whereas 9889 included only partners, 23,177 included only mother-offspring dyads, and 4157 included only father-offspring dyads.

## Measures

We used beta weights from large, publicly available up-to-date genome-wide association studies listed the Supplementary Note 8. None of the used genome-wide association studies used data from MoBa. Polygenic indices were calculated using LDPred v.1[67], a Bayesian approach that uses a prior on the expected polygenicity of a trait (assumed fraction of non-zero effect markers) and adjusts for linkage disequilibrium (LD) based on a reference panel to compute SNPs weights. Genotypes were coordinated with the summary statistics, with the number of overlapping SNPs reported in Supplementary Note 8. LD adjustment was performed using the European subsample of the 1000 Genomes genotype data as LD reference panel[68]. The weights were estimated based on the heritability explained by the markers in the GWAS summary statistics and the assumed fraction of markers with non-zero effects. For each GWAS trait we created LDpred PGI with the −score command in plink2[69]. Prior to calculating correlations between partners and relatives, we residualised the polygenic indices by regressing out the first 20 principal components of genetic ancestry, as well as chip, imputation, and batch number.

## Statistics

The polygenic index correlations (and 95% confidence intervals) were attained by correlating the residualised polygenic indices between partners and relatives using *cor.test* in R[70] 4.0.3. We tested whether the observed correlations were consistent with equilibrium by fitting structural equation models to data on mother-father-child trios, and testing whether a model constrained to equilibrium via equal variance across generations resulted in significantly worse fit. These models were estimated using OpenMx[71] 2.20.6. We describe this procedure in more detail in the Supplementary Note 6.

## Reporting summary

Further information on research design is available in the Nature Portfolio Reporting Summary linked to this article.

## Data availability

Data from the Norwegian Mother, Father and Child Cohort Study (MoBa) used in this study are managed by the national health register holders in Norway (Norwegian Institute of Public Health) and can be made available to researchers, provided approval from the Regional Committees for Medical and Health Research Ethics (REC), compliance with the EU General Data Protection Regulation (GDPR) and approval from the data owners. The consent given by the participants does not open for storage of data on an individual level in repositories or journals. Researchers who want access to data sets for replication should apply through helsedata.no. Access to data sets requires approval from The Regional Committee for Medical and Health Research Ethics in Norway and an agreement with MoBa.

## Code availability

Scripts used for simulations are provided in Supplementary Software 1 and at https://osf.io/dgw4r/. The summary statistics and reproducible code for the figures in this manuscript are also available at https://osf.io/dgw4r/.

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

## Acknowledgements

This work is part of the REMENTA and PARMENT projects and was supported by the Research Council of Norway (#300668 and #334093, respectively, to F.A.T.). The data acquisition, project management, and researcher positions were supported by the Research Council of Norway (#262177 and #336078 to E.Y., in addition to #288083). E.Y. is funded by the European Union (Grant agreement #101045526 and #818425). Views and opinions expressed are however those of the authors only and do not necessarily reflect those of the European Union or the European

Research Council Executive Agency. Neither the European Union nor the granting authority can be held responsible for them. E.C.C. is supported by the Research Council of Norway (#274611) and the South-Eastern Norway Regional Health Authority (#2021045). The Norwegian Mother, Father and Child Cohort Study is supported by the Norwegian Ministry of Health and Care Services and the Ministry of Education and Research. We are grateful to all the participating families in Norway who take part in this on-going cohort study. We thank the Norwegian Institute of Public Health (NIPH) for generating high-quality genomic data. This research is part of the HARVEST collaboration, supported by the Research Council of Norway (#229624). We also thank the NORMENT Centre for providing genotype data, funded by the Research Council of Norway (#223273), South East Norway Health Authorities and Stiftelsen Kristian Gerhard Jebsen. We further thank the Center for Diabetes Research, the University of Bergen for providing genotype funded by the ERC AdG project SELECTionPREDISPOSED, Stiftelsen Kristian Gerhard Jebsen, Trond Mohn Foundation, the Research Council of Norway, the Novo Nordisk Foundation, the University of Bergen, and the Western Norway Health Authorities. This work was performed on the TSD (Tjeneste for Sensitive Data) facilities, owned by the University of Oslo, operated and developed by the TSD service group at the University of Oslo, IT-Department (USIT). This work was partly supported by the Research Council of Norway through its Centres of Excellence funding scheme, (#262700).

## Author contributions

H.F.S. conceived of the idea, designed the theoretical models, and derived the relevant equations. E.C.C., R.C. and A.C.S. contributed to sample preparation and quality control of genomic data and polygenic indices with support from E.Y. H.F.S. carried out the analyses with support from N.H.E. and R.C. H.F.S. planned and carried out the simulations with help from E.M.E. and F.A.T. T.H.K contributed to the interpretation of the results. H.F.S. wrote the manuscript (incl. the supplementary information) with input from all authors. E.C.C. wrote parts of the methods. E.Y. and E.C.C. contributed to data generation and acquisition. E.M.E. and F.A.T. supervised the project. All authors provided critical feedback, discussed the results, and helped shape the manuscript.

## Competing interests

The authors declare no competing interests.
