## [Peer Review File · Nature Communications]

Genetic similarity between relatives provides evidence on the presence and history of assortative matingReviewer #1 (Remarks to the Author):

REVIEW

In this study, Sunde et al. explore how genetic similarity at a trait-associated loci can inform the nature and dynamics of assortative mating (AM) in the Norwegian population. The study contains both theoretical and empirical data for up to 15 traits. Overall, I really enjoyed reading this manuscript, which is written with great clarity. My comments are meant to improve its overall quality.

MAJOR COMMENTS

Novelty.

I could not help but noticing that the theoretical part of the manuscript refers to known and previously established concepts. The authors also recognize this themselves. Therefore, it is unclear to me why they decided to underline these results here again (although they are doing a great job at explaining them). Besides, previous studies have also used correlations of polygenic scores (PGS, this is the terminology that I use) to inform assortative mating (e.g., Okbay et al Nat Genet. 2022; Balbona et al. Behav genet. 2021, Torvik et al. Nat. Comm. 2022). I'm only underlying some of the references already cited in this study) but there is more.

One of the big challenges in AM research is quantifying its dynamics, which is one the goals of this paper. This is the part that I found the most interesting but here too, the authors mostly rely on a previously developed approach (by some of the same authors; Torvik et al. Nat. Comm. 2022) and apply it to the same cohort to conclude that height and BMI are in AM-equilibrium but psychological traits were not.

Inconsistent results.

I was intrigued by the inconsistency regarding whether the population assortatively mating on educational attainment (EA) has reached an equilibrium or not. The authors attribute this discrepancy to statistical power but this has not been clearly demonstrated.

Another interesting discrepancy is the fact that the authors do not replicate the evidence of AM on the genetic liability to depression, which was an important result in Torvik et al. (2022). This is not even discussed in this paper yet begs a few questions. What is different between their previous subsample and the one used here? Is there any evidence of differential ascertainment (or overlap with GWAS data) between those samples? I suggest that the authors give this a bit more consideration.

MINOR COMMENTS

- Why did the authors focus their disequilibrium analysis on trios only? I thought the model from Torvik et al. (2022) could accommodate any type of relationship? I'm wondering if the authors haven't made the ascertainment worse than it is by selecting trios. Also, they might have ignored valuable information.
- Can the authors provide tables with the correlations (and standard errors) reported in each Figure?
- If the available then it would help to see what is the accuracy of the different PGS in the MoBA sample.
- there were different selections for trios and it is not clear which set of trios was used to generate the results presented in the main text

Reviewer #2 (Remarks to the Author):

Sunde et al. examine the genetic consequences of assortative mating (AM) in terms of expected and observed correlations between relatives' genetic liabilities. Specifically, they first re-derive previous results regarding the expected genetic correlations under univariate assortative mating across relative

classes from a structural modeling perspective. They then use estimated polygenic indices (PGI) to estimate genetic correlations between partners and across relative classes, which they then compare to theoretical expectations. They also use estimated PGI to examine evidence for intergenerational equilibrium. Particular strengths of the paper include:

- the authors are careful not to overinterpret their results given their reliance on estimated PGI
- the authors' approach to examining deviations from equilibrium don't require that the PGI are high quality

There are two major limitations that the authors have not yet addressed: under AM, estimated PGI aren't only noisy but biased (Border 2023, Science) and they assume a simple model of univariate AM.

1. AM biases PGI

This is a fact that has been overlooked by many investigators using estimated PGI to measure AM, or more broadly, that AM biases statistical genetic estimators beyond the changes it induces in genetic architecture. In short, under AM, each causal variant tags every other causal variant, leading to bias in GWAS estimates. These minor biases compound when estimating any correlations (e.g., cross-mate, cross-relative, etc) involving estimated PGI. Crucially, this bias will only occur when AM is actually going on, so it won't lead to false positive evidence for AM, but it does substantially complicate inference. The authors need to discuss this limitation at the very least, and ideally would perform simulations to see how this impacts their results.

2. Assumptions of univariate AM and lack of transmitted environment

The authors' computations assume univariate AM, despite substantial evidence to the contrary (e.g., across height and educational attainment). Multivariate AM will both affect genetic architecture and exacerbate bias in PGI. Again, the authors need to discuss this limitation at the very least, and ideally would address this theoretically. Further, though they mention this limitation in the discussion, the assumption of no shared or transmitted environment (and the resulting gene-environment correlations it would induce under AM) is undoubtedly wrong for many traits of interest. E.g., educational attainment is affected by financial circumstances, which are also transmitted, so education PGI tag environmental components of SES. Given the strength of these assumptions, it doesn't seem particularly useful to estimate, e.g., the estimated number of generations of stable assortative mating for a particular trait.

Overall, the manuscript is well considered but uses a simplistic model. This model is still far less simplistic than the random mating model that dominates the statistical genetics literature, and the observation of deviation from equilibrium is valuable to the field. I personally think the theoretical results and path tracing stuff should go in the supplement as the results aren't new. Additionally, I'd find a simple table illustrating the expected phenotypic and genetic correlations across relationship classes for 1, 3, 5 generations and equilibrium as a function of parameters of interest far more helpful than the path diagram in Figure 1, which could be moved to the supplement. But this is a matter of preference.

Response to reviewers

We have made major revisions to our manuscript in response to the reviewers' comments. The largest change is in the supplementary information, which is now completely overhauled. It now includes advanced theoretical models – both for equilibrium and disequilibrium – with environmental transmission and other sources of familial resemblance. Furthermore, we have added realistic simulations, which validate our theoretical expectations both for the true genotypic correlations and the correlations between polygenic indices. We have updated the manuscript to reflect these changes.

All the changes in the main manuscript relevant to reviewers' comments are mentioned below next to the relevant comment. We have also made minor changes to improve clarity beyond what was commented on by reviewers, as well as changes required by formatting guidelines (e.g., removed subheadings in the discussion). All of these are clearly marked in the revised manuscript. Finally, we included the polygenic index for drinking and recalculated the polygenic index for BMI (we discovered a mistake with the original calculation of this index). We have changed the text in relevant places to reflect the new results (BMI is no longer in equilibrium).

Reviewer #1's comment	Our response and relevant changes
In this study, Sunde et al. explore how genetic similarity at a trait-associated loci can inform the nature and dynamics of assortative mating (AM) in the Norwegian population. The study contains both theoretical and empirical data for up to 15 traits. Overall, I really enjoyed reading this manuscript, which is written with great clarity. My comments are meant to improve its overall quality.	We thank the reviewer for their thorough and helpful comments, which we address below.

Novelty. I could not help but noticing that the theoretical part of the manuscript refers to known and previously established concepts. The authors also recognize this themselves. Therefore, it is unclear to me why they decided to underline these results here again (although they are doing a great job at explaining them).	Much of the theoretical results are indeed previously established. However, we found many of the previous sources wanting in clarity. This highlights the need for a clearer explanation (which, given reviewer #1's comment, we believe we have achieved. Numerous comments we have received from readers of the preprint attest to this as well). However, lack of clarity is not the only reason for redoing the work. The key motivation for redoing the work with path analysis is that path analysis offers a ready way to relax assumptions, such as asking "what happens to imperfect polygenic indices?" or (in our revised manuscript) "what happens if there are environmental transmission as well?". These questions are not answered sufficiently in earlier work and requires rederiving the theoretical results. We have revised the final paragraph in the introduction to highlight the motivation to rederive the theoretical results: Line 90 (p.4): In this paper, we study the extent of assortative mating on a range of phenotypes and its historical consequences by using genetic data from extended family members. Our first aim is to derive the expected genotypic correlations between family members under various assumptions using path analysis. There are earlier theoretical papers that lays out the consequences of assortative mating on familial resemblance^{1-5,30}. However: 1) they often focus on phenotypic rather than genotypic resemblance; 2) they don't consider imperfectly measured genetic factors (i.e., polygenic indices); 3) they often do not consider gene-environment correlations; and 4) they either don't consider disequilibrium or do so only under simplistic assumptions. We use path analysis because it offers a ready way to relax assumptions while making the theory accessible for non-specialists. The path analysis in the main text serves, in part, as an introduction to the supplementary information, where we relax assumptions to see what happens to the correlations. At the end of the first paragraph in the results, we have added the following sentence: Line 125 (p.5): Path diagrams with relaxed assumptions (e.g., gene-environment correlations) are presented and discussed in the supplementary information, where we also provide simulations validating our theoretical expectations.
Besides, previous studies have also used correlations of polygenic scores (PGS, this is the terminology that I use) to inform assortative mating (e.g., Okbay et al Nat Genet. 2022; Balbona et al. Behav genet. 2021, Torvik et al. Nat. Comm. 2022). I'm only underlying some of	Previous studies have indeed investigated correlations of polygenic indices, although they have largely limited themselves to looking at partners rather than relatives. Furthermore, they have either been limited by smaller sample sizes (and by extension statistical power), or they have "guessed" who is partnered with whom based on proxies of unknown reliability (e.g., living in roughly the same place). Here, we have the largest number of partners who are both confirmed partners and have been genotyped. We have emphasised this in the introduction:

the references already cited in this study) but there is more.	Line 73 (p.3): [...] but lack of statistical power left the question unresolved for most traits. Here, we remedy this by investigating partners in the Norwegian Mother, Father, and Child Cohort Study (MoBa)^{26,27}, the largest cohort of confirmed partners with available genetic data (n=47,135).
One of the big challenges in AM research is quantifying its dynamics, which is one the goals of this paper. This is the part that I found the most interesting but here too, the authors mostly rely on a previously developed approach (by some of the same authors; Torvik et al. Nat. Comm. 2022) and apply it to the same cohort to conclude that height and BMI are in AM-equilibrium but psychological traits were not.	There are several key differences between the approach we use in this paper and the approach developed in Torvik 2022. We have made changes to make this clearer to readers (see responses to the next two comments). In short, Torvik 2022 used a structural equation model that required both phenotypic data and polygenic indices to work, whereas here we only look at polygenic indices. Furthermore, while we use path analysis for theoretical expectations, the empirical results are basically limited to correlations between family members and differences in variances across generations. The approach developed in Torvik 2022 is powerful but requires assumptions about the causes of phenotypic partner similarity that we do not have to make here.
Inconsistent results. I was intrigued by the inconsistency regarding whether the population assortatively mating on educational attainment (EA) has reached an equilibrium or not. The authors attribute this discrepancy to statistical power but this has not been clearly demonstrated.	We have added the following remarks in the discussion: Line 326 (p.14): [...]. This would also explain the discrepancy between our conclusion and that in Torvik, et al.¹⁹, who found no significant deviation from equilibrium using an earlier version of data from the same cohort. We primarily used variance differences across generations whereas Torvik, et al.¹⁹ compared the predicted and expected correlations between siblings and partners. Considering that the sibling correlation in Fig. 3B is significantly lower than expected given equilibrium, the change in result likely stem from an increase in statistical power, owing to more genotyped individuals available in the current sample and the use of parent-offspring dyads instead of sibling dyads.
Another interesting discrepancy is the fact that the authors do not replicate the evidence of AM on the genetic liability to depression, which was an important result in Torvik et al. (2022). This is not even discussed in this paper yet begs a few questions. What is different between their previous subsample and the one used here? Is there any evidence of differential ascertainment (or overlap with GWAS data) between those	This is indeed interesting on first reading, but the explanation is unfortunately rather mundane. It can be directly attributed to the different approach taken in Torvik 2022, which leverages phenotypic data in a structural equation model to find evidence of assortative mating. The empirical correlations between polygenic indices reported in Torvik 2022 is consistent with those reported here (as they should be considering Torvik 2022 used a subsample of the sample used in the current paper). As for ascertainment, we believe looking at parent-offspring correlations are less ascertained than sibling correlations, which we will explain in our response to the next comment. Our manuscript now includes a discussion on the discrepancy between our finding and that in Torvik 2022:

samples? I suggest that the authors give this a bit more consideration.	Line 281 (p.12): Our findings seemingly contradict Torvik, et al. ¹⁹, who reported evidence of assortative mating on depression using a smaller sample of the same cohort. However, that paper used a structural equation model requiring both genetic and phenotypic data, and the polygenic index correlations reported in that paper match those we report here. This could indicate that phenotypic partner similarity in mental health is caused by processes other than assortative mating, such as convergence²⁰ (which was not modelled in Torvik, et al. ¹⁹). On the other hand, they could also be false negatives resulting from inadequate polygenic indices. We have also added the following to the methods-section, underlying that there was no sample overlap with the GWAS-discovery samples: Line 416 (p.17): None of the used genome-wide association studies used data from MoBa.
Why did the authors focus their disequilibrium analysis on trios only? I thought the model from Torvik et al. (2022) could accommodate any type of relationship? I'm wondering if the authors haven't made the ascertainment worse than it is by selecting trios. Also, they might have ignored valuable information.	As mentioned, the model from Torvik 2022 requires phenotypic data, which we do not use. We focused on trios because (1) we needed to include partners (i.e., mother-father dyads) regardless, and (2) there are many more parent-offspring dyads in the data than other types of dyads. Even though ascertainment is an ever-present issue in studies like this, we believe parent-offspring dyads is less affected than other dyads because the cohort itself is inherently selecting on mother-offspring dyads, with strong recruitment of fathers as well. Siblings are available in the data because both parties enrolled or mothers enrolled twice (i.e., doubly ascertained), whereas parent-offspring correlations are available in the data because mothers enrolled (at least) once (i.e., singly ascertained). Obviously, there may be some ascertainment bias for the partner correlation (which requires data on both the mother and father), but there is not much we can do about that. Additionally, as we show in the supplementary, using more distant relatives require stronger assumptions about the quality of the polygenic index and the absence of gene-environment correlation – assumptions which are not required within the nuclear family. We have adapted the relevant paragraph in the methods section, where we now make clear why we focus on trios. Line 401 (p.16): To test equilibrium, we used all available mother-father-child trios from MoBa. We relied on trios to test equilibrium for the following reasons: 1) It allowed estimating the partner correlation and parent-offspring correlations in the same model; 2) it allowed us to include both the mother-offspring and father-offspring simultaneously thus increasing statistical power; 3) it allowed us to estimate variances separately for the two generations; 4) there was no need to distinguish between correlations between relatives in the parent generation and in the offspring generation, as this is inherent in the design; 5) focusing on the

	nuclear family removes the need to make assumptions about the genetic signal or gene-environment correlations; and finally, 6) the sample in MoBa is inherently selected on parent-offspring dyads whereas the availability of other relatives is coincidental. Using other relatives, such as siblings, could therefore lead to stronger ascertainment bias. After randomly selecting one offspring from each nuclear family, we were able to construct a sample of 81,145 genotyped mothers, 54,550 genotyped fathers, and 71,544 genotyped offspring, resulting in a total of 93,767 incomplete and complete trios. Of these, 36,764 were complete trios, whereas 10,371 included only partners, 24,659 included only mother-offspring dyads, and 4,914 included only father-offspring dyads.
Can the authors provide tables with the correlations (and standard errors) reported in each Figure?	Yes. The requested table (Supplementary Table S17) is now in the supplementary information, section 7. (albeit with 95% confidence intervals instead of standard errors). As in the original submission, the correlations are also available as .rds-files from the link provided under each figure.
If the available then it would help to see what is the accuracy of the different PGS in the MoBA sample.	This is not available. Also, the accuracy of the different polygenic indices will only be informative if one is assuming direct phenotypic assortative mating, something our approach does not require. Our results are agnostic as to what the polygenic index actually measures, and what related phenotype(s) are assorted upon. We have made this clearer by adapting the final limitation. This change is also motivated by comments made by reviewer #2 (see below). Line 374 (p.15): Third, the approach we use here is agnostic as to which trait(s) the polygenic indices actually measure, and which phenotype(s) are being assorted upon. Future research may want to investigate what set of phenotypes mediate the polygenic index correlations between partners, as it may not always be attributable to the phenotype that the polygenic index supposedly measures.
there were different selections for trios and it is not clear which set of trios was used to generate the results presented in the main text	We have clarified why we used trios and how they were selected in a previous response. There were not any different selections for trios.

Reviewer #2's comment	Our response and relevant changes
Sunde et al. examine the genetic consequences of assortative mating (AM) in terms of expected and observed correlations between relatives' genetic liabilities. Specifically, they first re-derive previous results regarding the expected genetic correlations under univariate assortative mating across relative classes from a structural modeling perspective. They then use estimated polygenic indices (PGI) to estimate genetic correlations between partners and across relative classes, which they then compare to theoretical expectations. They also use estimated PGI to examine evidence for intergenerational equilibrium. Particular strengths of the paper include:  - the authors are careful not to overinterpret their results given their reliance on estimated PGI - the authors' approach to examining deviations from equilibrium don't require that the PGI are high quality There are two major limitations that the authors have not yet addressed: under AM, estimated PGI aren't only noisy but biased (Border 2023, Science) and they assume a simple model of univariate AM.	We thank the reviewer for their thorough and helpful comments, which we address below.

1. AM biases PGI

This is a fact that has been overlooked by many investigators using estimated PGI to measure AM, or more broadly, that AM biases statistical genetic estimators beyond the changes it induces in genetic architecture. In short, under AM, each causal variant tags every other causal variant, leading to bias in GWAS estimates. These minor biases compound when estimating any correlations (e.g., cross-mate, cross-relative, etc) involving estimated PGI. Crucially, this bias will only occur when AM is actually going on, so it won't lead to false positive evidence for AM, but it does substantially complicate inference. The authors need to discuss this limitation at the very least, and ideally would perform simulations to see how this impacts their results.

We agree that this complicates inference but disagree that this is a *major* limitation: While such cross-tagging of causal variants does indeed increase the polygenic index correlation between partners, we do not believe it is a bias per se: The *true* polygenic index correlation will and should include the covariance between different loci. We think the key thing to communicate to readers (given Reviewer #2's comment) is that the genetic signal (i.e., squared correlation between polygenic index and true genetic factor) is *not* the degree to which the polygenic index weights match the true direct effects, and is as such not a measure of the accuracy of the polygenic index weights. The covariance between different loci will affect all relatives equally and is therefore not a problem for the approach we use here. However, we agree that we should be more careful with our inferences and have therefore made the following changes.

We have first and foremost heavily expanded our treatment of polygenic indices in the supplementary information (Sections 3, 4.4, and 5.4), and have included simulations that, among other things, show that the genetic signal (the squared correlation between the polygenic index and the true genetic factor) increases under assortative mating, which in turn increases the polygenic index correlations. However, the polygenic index correlations between partners and relatives still conform to our expectations, meaning our substantive results does not depend on this assumption.

We have also made the following changes in the main text:

In the introduction, we have revised the wording to make it clear that the induced correlations are not only between the causes of the assorted trait, but whatever is associated with the trait:

Line 68 (p.3):

If partner similarity arises because of assortative mating, then this will induce cross-partner correlations between factors that are associated with the trait.

In the results, we have added the following sentence to highlight that the correlation between true genetic factor and polygenic index is increased by this “cross-tagging of variants”, and thereby is not simply the *accuracy* of the polygenic weights.

Line 152 (p.6):

Assortative mating will induce covariance between different loci (i.e., linkage disequilibrium), which is included in the genetic signal. This means that s may be larger than the correlation between the true direct effects and the polygenic index weight, and as such do not represent the accuracy of the polygenic index weights (see Supplementary Information, section 3 and section 4.4).

We have also made a small correction to the expected polygenic index between distant relatives because we noticed that we had made a minor mistake before we received the reviews. We discovered that the polygenic index correlation between distant

relatives can only be approximated unless the genetic signal is known (Although the inaccuracy is in most cases negligible, see supplementary information, section 3.1). The discrepancy was too small to notice in the simulations we ran before first submission.

Line 173, (p.8):

As for polygenic index correlations, they can be approximated by replacing ρ_g with ρ_{pgi} in equation 5, although depending on the genetic signal, the true polygenic index correlation may be slightly higher (see supplementary information, section 3).

In the discussion, we have made two changes. First, we made a small change to an existing sentence warning that the partner correlation only informs about the presence of assortative mating, not its degree (similar to what the reviewer's comment highlights), and appended a sentence highlighting that the genetic signal is itself a function of assortative mating:

Line 294 (p.13):

A corollary is that the correlations reported here do not quantify the exact degree of assortative mating because it is confounded by the genetic signal of the polygenic index. Complicating inference further is that the genetic signal is itself increased under assortative mating.

Second, we adapted the final limitation to highlight that our study remains agnostic as to what the polygenic indices are measuring. This change is also motivated by the next comment about assuming univariate AM.

Line 374 (p.15):

Third, the approach we use here is agnostic as to which trait(s) the polygenic indices actually measure, and which phenotype(s) are being assorted upon. Future research may want to investigate what set of phenotypes mediate the polygenic index correlations between partners, as it may not always be attributable to the phenotype that the polygenic index supposedly measures.

2. Assumptions of univariate AM and lack of transmitted environment

The authors' computations assume univariate AM, despite substantial evidence to the contrary (e.g., across height and educational attainment). Multivariate AM will both affect genetic architecture and exacerbate bias in PGI. Again, the authors need to discuss this limitation at the very least, and ideally would address this theoretically. Further, though they mention this limitation in the discussion, the assumption of no shared or transmitted environment (and the resulting gene-environment correlations it would induce under AM) is undoubtedly wrong for many traits of interest. E.g., educational attainment is affected by financial circumstances, which are also transmitted, so education PGI tag environmental components of SES. Given the strength of these assumptions, it doesn't seem particularly useful to estimate, e.g., the estimated number of generations of stable assortative mating for a particular trait.

This comment can be divided into two: Univariate AM and lack of transmitted environment. We have put most of our efforts addressing the second of these comments and will therefore respond to that first:

Lack of transmitted environment:

We have completely overhauled the supplementary information to address this comment, along with appropriate changes in the main paper. We were originally working on a theoretical follow-up paper where we would expand the simplistic model presented here to include, among other things, transmitted environment, supplemented with simulations. We have decided to merge this into the supplementary information. (This is why it also contains things that were not commented upon by either reviewer, such as dominance effects, which we have kept as it may be of interest to some readers).

In short, the supplementary information now includes sections where the model in Fig.1 is expanded to include dominance effects, sibling-shared environmental effects, cultural transmission (environment → environment) and phenotypic transmission (phenotype → environment). Furthermore, we detail how environmental transmission (and the resulting gene-environment correlations) changes the dynamics under disequilibrium, and how correlations between partners' and relatives' polygenic indices are affected. We also provide accurate simulations that validate the theoretical expectations. In short, gene-environment correlations will mimic higher heritability, and thereby increase the genetic consequences of assortative mating. (Another way to think about it is that the genotype-phenotypic correlation is what really matters, not the heritability per se). The relationship between the polygenic index correlation between partners and between first-degree relatives remain the same as before (meaning those correlations are not dependent on assumptions about gene-environment correlations). The correlation between distant relatives, however, will be slightly higher than expected given the algorithm presented in equation 5: $\left(\frac{1+\rho_g}{2}\right)^k$.

We have also made the following changes in the main text:

In the introduction, we have added gene-environment correlations as a motivation for redoing the theoretical derivations (although it is included as a result of this revision):

Line 95, (p.4):

There are earlier theoretical papers that lays out the consequences of assortative mating on familial resemblance^{1-5,30}. However, [...], 4) they often do not consider gene-environment correlations, and 5) they either don't consider disequilibrium or do so only under simplistic assumptions.

In the results, at the end of the first paragraph, we have added the following sentence:

Line 125 (p.5):

Path diagrams with relaxed assumptions (e.g., gene-environment correlations) are presented and discussed in the supplementary information, where we also provide simulations validating our theoretical expectations.

In the results, we have included the following paragraphs along with their own subheading:

Line 191 (p.8):

Gene-environment correlations, shared environment, and dominance effects

One limitation with most earlier work, such as Fisher¹, is that they assume a simplistic model where genetic similarity is the only cause of familial resemblance. In the Supplementary Information, we detail how genetic similarity between relatives are affected by dominance effects, shared environmental effects, and various forms of environmental transmission. If genetic and environmental transmission occur simultaneously, assortative mating will induce (and greatly increase) correlations between genetic and environmental factors. Such gene-environment correlations will, in this context, mimic higher heritability, leading to higher genotypic correlations between partners and thereby exacerbated genetic consequences of assortative mating. However, the relationship between the genotypic correlation between partners and the genotypic correlation between close/first-degree relatives will stay the same, meaning Equation (2) and Equation (4) can be used without making assumptions about gene-environment correlations or other sources of familial resemblance.

This is not the case for distant relatives. If there are substantial shared environmental effects, gene-environment correlations, or other sources of familial resemblance, the properties of Fig. 1 that allow the general algorithm in Equation (5) are no longer present. This is because non-genetic causes of familial resemblance result in pathways between distant relatives that bypass the genotypes of intermediate relatives, thus increasing the true genotypic correlation to beyond what Equation (5) would predict. Equation (5) still serves as a rough approximation, although any statistical model that relies on it could be biased if such extra pathways exist.

In the discussion, we have added the following sentence at the end of the paragraph about theoretical expectations:

Line 270 (p.12):

We also described how assortative mating can induce and increase gene-environment correlations, which mimic higher heritability and thereby exacerbate the genetic consequences of assortative mating – especially correlations between distance relatives.

We have also added the following paragraph discussing how our results are consistent with gene-environment correlations for educational attainment:

Line 307 (p.13):

For educational attainment, the polygenic index correlations between first-degree relatives are lower than expected (indicating disequilibrium, see below) while correlations between third- and fourth-degree relatives are higher than expected. This is consistent with substantial gene-environment correlations for educational attainment⁴⁴⁻⁴⁶. In the supplementary information, we showed theoretically and with simulations that correlations between higher-degree relatives (but not first-degree relatives) will be higher than expected given Equation (5) if such gene-environment correlations are present.

Univariate AM:

We believe the current understanding of multivariate assortative mating is lacking and are therefore working on another paper where we attempt to theoretically clarify what it is and what is its consequences. We therefore agree that multivariate assortative mating must be addressed theoretically. However, we believe that doing so here is neither necessary nor suitable. First and foremost, we do not believe this is a *major* limitation for the current paper because we are not assuming that the assorted phenotype (P in the path diagrams) must be the phenotype that the polygenic index supposedly measures. That is why we, for example, write “The phenotype that is assorted on is denoted with P_{it} ”, which we have carefully formulated to avoid making assumptions about type of assortative mating. The assorted phenotype can be, and probably is, best construed as consisting of *the set of traits* that partners are assorting on (e.g., a composite of, say, height and educational attainment). However, we feel that a lengthy discussion elaborating and defending this view would not be suitable in this paper because: (1) It requires its own paper (which we are working on), (2) it would take up too much space (the supplementary information is long enough as it is), (3) it would distract from the main purpose of this paper, and (4) the conclusions of this paper do not depend on type of assortative mating, which in turn means it does not strongly depend on assuming univariate assortative mating.

We have, however, made changes to highlight this limitation more prominently (under future directions):

Line 374 (p.15):

Third, the approach we use here is agnostic as to which trait(s) the polygenic indices actually measure, and which phenotype(s) are being assorted upon. Future research may want to investigate what set of phenotypes mediate the polygenic index correlations between partners, as it may not always be attributable to the phenotype that the polygenic index supposedly measures.

Overall, the manuscript is well considered but uses a simplistic model. This model is still far less simplistic than the random mating model that dominates the statistical genetics literature, and the observation of deviation from equilibrium is valuable to the field. I personally think the theoretical results and path tracing stuff should go in the supplement as the results aren't new. Additionally, I'd find a simple table illustrating the expected phenotypic and genetic correlations across relationship classes for 1, 3, 5 generations and equilibrium as a function of parameters of interest far more helpful than the path diagram in Figure 1, which could be moved to the supplement. But this is a matter of preference.

We agree that the models used in the initial submission can be characterised as simplistic. We believe this is no longer the case, now that we also use models that incorporate environmental transmission and gene-environment correlations. We hope Reviewer #2 agrees.

As for whether to move stuff to the supplementary: We appreciate this comment but have for the time being elected to keep much of the path tracing in the main text rather than just summarizing the equations. It is, as the reviewer says, a matter of preference. We have elected to keep it as is because the knowledge we wish to communicate is not in the equations themselves, but in *why* the equations are what they are. This is often lost in prior work (some of which are infamously inaccessible) and would be lost here too if we merely listed the equations. Path diagrams also offers a ready way for curious readers to relax assumptions themselves should they wish to do so (i.e., add more paths), something that original sources do not readily invite. We have received comments from several readers of the preprint that they found our explanation very helpful, so there is a clear market for this even though much of it is technically known.

As for a table for correlations after 1, 3, 5, and ∞ generations: As far as we can tell, there are no closed-form equation for genetic similarity in disequilibrium that does not involve iterating the same set of equations starting from the base population, making such a table impossible. We could of course calculate it for various example values, although we are not sure this would be as helpful.

Reviewer #1 (Remarks to the Author):

I thank the authors for addressing each of my (and the other reviewer's) comments. The revised manuscript has considerably improved consequently.

The only point that is left somewhat unsatisfactory is whether there is evidence of AM depression. This is an important question, which this paper reopens without giving enough guidance for what could be the answer. I understand Torvik et al. (2022) used phenotypic information and therefore might be capturing more genetic signal (if there is truly AM) than just using the PGS. The other explanation, i.e. convergence, is also likely to explain the discrepancy. However, I thought that having some measure of the prediction accuracy could help resolve if not hint at what could be the answer (lack of power or not AM at all, e.g., convergence). I thought the author had a depression score available in MoBa and therefore could quantify the accuracy of their predictor.

I don't feel strongly (so the authors may decide not to do that) but I believe that going an extra half-mile and assessing the accuracy of their scores (at least for height, BMI and depression) would really help. My intuition is that their depression score is not very predictive, which could give a clearer direction to follow for future research.

Reviewer #2 (Remarks to the Author):

I still have substantial concerns about the manuscript, though these concerns can be remedied through future revision.

1. The authors have misunderstood stood my point about biased PGI under AM, writing "While such cross-tagging of causal variants does indeed increase the polygenic index correlation between partners, we do not believe it is a bias per se: The true polygenic index correlation will and should include the covariance between different loci."

It is correct that the true PGI correlation will and should include the covariance between different loci. But the estimated PGI overestimates this quantity as well. This is proven in the paper referenced in my first review (page 14 of supplement of Border et al 2023) for the case of even odd polygenic scores but it's simple to see in other cases: Under AM, the i,j th offdiagonal element of the LD correlation matrix W is proportional to $\beta_i \beta_j$ so the LD matrix can be approximated (ignoring local LD) as

$$W = D + a \beta \beta^T$$

where D is diagonal, a is a scalar, and β is a vector. So, e.g., the true genetic variance will be larger reflecting this increase in covariance:

$$\beta^T W \beta = \beta^T \beta + a (\beta^T \beta)^2$$

But with PGIs, we don't use β , we use estimates b from GWAS. Assuming everything is standardized, the j th GWAS slope is

$$E[b_j] = \beta_j + \sum_{i \neq j} W_{i,j} \beta_i$$

and the PGI is then

$$Xb = X W \beta$$

such that the variance of PGI is

$$b^T X X b = \text{beta.T W W W beta}$$

which is not beta.T W beta , the true (though inflated by AM induced LD) genetic variance.

In the case of cross-relative PGI correlations, the middle of the three Ws will be a cross-relative cross-LD matrix (say R), so a bit different, and the two outer Ws will still be there.

This is bias above and beyond the true PGI correlation. You want to estimate beta.T R beta but you instead have beta.T W R W beta . So my first major criticism remains unaddressed but my recommendations remain the same:

The authors need to discuss this limitation at the very least, and ideally would perform simulations to see how this impacts their results.

2. I am satisfied with the discussion of environmental transmission and G-E correlation--this makes the paper stronger. I also understand that a complex discussion of univariate vs multivariate AM is beyond scope. But I do think this is a limitation worth mentioning as they do assume that AM is unidimensional (though possibly involving multiple phenotypes). They write in their response:

"First and foremost, we do not believe this is a major limitation for the current paper because we are not assuming that the assorted phenotype (P in the path diagrams) must be the phenotype that the polygenic index supposedly measures. That is why we, for example, write 'The phenotype that is assorted on is denoted with *Pit*', which we have carefully formulated to avoid making assumptions about type of assortative mating. The assorted phenotype can be, and probably is, best construed as consisting of the set of traits that partners are assorting on (e.g., a composite of, say, height and educational attainment)."

You are still assuming that there is a single dimension being assorted on (even if it is a composite of multiple phenotypes) and not more than one. Seeing as there are strong cross-mate cross-trait correlations across so many phenotypes, e.g. education, height, smoking, alcohol, wealth, psychiatric disorders, etc., it's hard to imagine that these can all be reduced to a single dimension. Who ends up with whom is almost certainly complex.

So despite the increased complexity of the theoretical model, it is still simplistic in some important senses, which is fine, but these limitations need to be acknowledged explicitly. Further, statements about the estimated number of generations of stable assortment need to acknowledge that they rely on these assumptions.

Overall, I believe my issues can be addressed with further revision.

Response to reviewers (round 2)

We have made several revisions to our manuscript in response to the reviewers' comments, and added a new simulation in the supplementary information. We believe the manuscript has become greatly improved as a result and want to thank both reviewers for their insightful and helpful comments. All the changes in the main manuscript relevant to reviewers' comments are mentioned below next to the relevant comment. There are also some minor changes elsewhere (clearly marked) to improve the flow of the paper.

Reviewer #1's comment	Our response and relevant changes
I thank the authors for addressing each of my (and the other reviewer's) comments. The revised manuscript has considerably improved consequently. The only point that is left somewhat unsatisfactory is whether there is evidence of AM depression. This is an important question, which this paper reopens without giving enough guidance for what could be the answer. I understand Torvik et al. (2022) used phenotypic information and therefore might be capturing more genetic signal (if there is truly AM) than just using the PGS. The other explanation, i.e. convergence, is also likely to explain the discrepancy. However, I thought that having some measure of the prediction accuracy could help resolve if not hint at what could be the answer (lack of power or not AM at all, e.g., convergence). I thought the author had a depression score available in MoBa and therefore could quantify the accuracy of their predictor. I don't feel strongly (so the authors may decide not to do that) but I believe that going an extra half-mile and assessing the accuracy of their scores (at least for height, BMI and depression) would really help. My intuition is that their depression score is not very predictive, which could give a clearer direction to follow for future research.	Thank you for helping us improve it. As for depression, we have revised further. We agree that the most likely explanation is that the depression score is not very predictive and have therefore referenced the accuracy of the score from another paper (using the same cohort). The paragraph now makes it clear that a false negative is the most likely explanation. Line 286 (p.12): [...]. This could indicate that phenotypic partner similarity in mental health is caused by processes other than assortative mating, such as convergence²⁰ (which was not modelled in Torvik, et al. ¹⁹). On the other hand, the results could also be false negatives resulting from low-quality polygenic indices. The depression polygenic index only correlates .11 with the phenotype in the current cohort¹⁹, meaning the expected partner correlation is only about $.11^2 \times .14 = .0017$ under direct assortment. A false negative is therefore highly likely. Reports of smaller but non-zero phenotypic correlations prior to partner formation suggests that both convergence and assortment play an important role^{43,44}. PS: the expected correlation under direct assortment (.0017) assumes the phenotypic correlation is .14, which is taken from the meta-analysis mentioned earlier in the paper.

Reviewer #2's comment

I still have substantial concerns about the manuscript, though these concerns can be remedied through future revision.

1. The authors have misunderstood stood my point about biased PGI under AM, writing "While such cross-tagging of causal variants does indeed increase the polygenic index correlation between partners, we do not believe it is a bias per se: The true polygenic index correlation will and should include the covariance between different loci."

It is correct that the true PGI correlation will and should include the covariance between different loci. But the estimated PGI overestimates this quantity as well. This is proven in the paper referenced in my first review (page 14 of supplement of Border et al 2023) for the case of even odd polygenic scores but it's simple to see in other cases: Under AM, the i,j -th offdiagonal element of the LD correlation matrix W is proportional to $\beta_i\beta_j$ so the LD matrix can be approximated (ignoring local LD) as:

$$W = D + a\beta\beta^T$$

where D is diagonal, a is a scalar, and β is a vector. So, e.g., the true genetic variance will be larger reflecting this increase in covariance:

$$\beta^T W \beta = \beta^T \beta + a(\beta^T \beta)^2$$

But with PGIs, we dont use β , we use estimates b from GWAS. Assuming everything is standardized, the j -th GWAS slope is:

$$\mathbb{E}[b_j] = \beta_j + \sum_{i \neq j} W_{i,j} \beta_i$$

and the PGI is then:

$$Xb = XW\beta$$

such that the variance of PGI is:

$$b^T X X b = \beta^T W W \beta$$

Our response and relevant changes

We thank the reviewer for their helpful comments, and for patiently taking to the time to explain our misunderstanding in more detail. We have added a simulation and some corresponding revisions where we attempt to address this point.

If we have understood correctly, the concern is that the AM-induced linkage disequilibrium affects the cross-relative PGI-correlations in two ways: (1) through the actual induced LD (which should be part of the true correlation), and (2) through affecting the weights that the PGI is based on (which could be considered bias). In other words, the AM-induced LD will have much larger effects on (estimated) PGI-correlations than true genotypic correlations.

We are already unable to interpret too much from the absolute magnitude of the correlations (e.g., *Line 297 (p.13)*: "[...] the correlations reported here do not quantify the exact degree of assortative mating because it is confounded by the genetic signal of the polygenic index"), so we are not too concerned that we are estimating (standardized variants of) $\beta^T W R W \beta$ instead of $\beta^T R \beta$. Nonetheless, we have made further revisions to make this more explicit (see below). The more important question is whether and how this has impacted our conclusions. For our conclusions to be impacted, this bias must either differentially affect correlations between close and distant relatives, or it must affect the relationship between the partner correlation and the correlation between relatives.

To recap, we have so far assumed that what matters is the correlation between the calculated PGI and the true genetic factor (which we denote s for signal). If the PGI is biased, we have implicitly assumed that this will enter the picture via this correlation, and that we can remain agnostic to whether the PGI is biased or not. We may be wrong in this assumption. To find out, we have made an addition to the simulations reported in the Supplementary Information (**Section 5.6, pages 44 to 46**). Here, we simulated several generations of AM (to induce LD among the simulated variants) and calculated new PGI-weights based on the correlation between each locus and the phenotype. These new PGI-weights should be biased, unlike the weights in the original simulations (which were just noisy). We then calculated new polygenic indices and checked whether the correlations between various family members conformed to the equations laid out earlier. We find that they still do, both in disequilibrium and equilibrium. We are therefore confident that, while the polygenic indices are probably biased, our conclusions should still stand.

which is not $\beta^T W \beta$, the true (though inflated by AM induced LD) genetic variance.

In the case of cross-relative PGI correlations, the middle of the three W s will be a cross-relative cross-LD matrix (say R), so a bit different, and the two outer W s will still be there.

This is bias above and beyond the true PGI correlation. You want to estimate $\beta^T R \beta$ but you instead have $\beta^T W R W \beta$. So my first major criticism remains unaddressed but my recommendations remain the same:

The authors need to discuss this limitation at the very least, and ideally would perform simulations to see how this impacts their results.

2. I am satisfied with the discussion of environmental transmission and G-E correlation--this makes the paper stronger. I also understand that a complex discussion of univariate vs multivariate AM is beyond scope. But I do think this is a limitation worth mentioning as they do assume that AM is unidimensional (though possibly involving multiple phenotypes). They write in their response:

"First and foremost, we do not believe this is a major limitation for the current paper because we are not assuming that the assorted phenotype (P in the path diagrams) must be the phenotype that the polygenic index supposedly measures. That is why we, for example, write "The phenotype that is assorted on is denoted with P_{it} ", which we have

In addition to the new section in the supplementary (Section 5.6, pages 44 to 46), we have made the following addition in the main manuscript (in the limitations-section of the discussion):

Line 362 (p.14):

[...] Furthermore, assortative mating can bias GWAS estimates and thereby bias polygenic indices⁶⁰. Although this should not affect our conclusions (see Supplementary Information, section 5.6), it does make it difficult to precisely quantify the strength of assortative mating on various traits and hence the magnitude of the genetic consequences.

We have also made some minor revisions (underlined) on page 24 of the Supplementary Information (Section 3.1):

First paragraph:

Note that many different causes of polygenic index inaccuracy (such as including loci without true effects, missing loci that have true effects, or overestimating the loci's true effects due to linkage disequilibrium) all amounts to the same thing: A difference between the loci's true effects on the phenotype and on the polygenic index.

Second paragraph:

[...]. This, in turn, means that the genetic signal – the shared variance between the polygenic index and the true genetic factor – is s^2 . Just like with h^2 (the heritability), s^2 will include the covariance between different loci (linkage disequilibrium) and is therefore not just the sum of the individual loci's effects. This is especially true for polygenic indices where the weights themselves are biased by linkage disequilibrium²¹.

Who ends up with whom is indeed almost certainly complex. In our revision, we have explicitly acknowledged the assumption of unidimensional assortment at the end of the manuscript.

Line 385 (p.15):

[...] Furthermore, we have assumed assortment is unidimensional. Considering ample evidence of partner correlations across different traits^{44,60}, future studies may want to extend this line of research to multidimensional assortment.

When it comes to the number of generations of assortment, we have made several minor revisions with the goal of being more explicit in what assumptions underlie this number and how to interpret it. We have attempted to more

carefully formulated to avoid making assumptions about type of assortative mating. The assorted phenotype can be, and probably is, best construed as consisting of the set of traits that partners are assorting on (e.g., a composite of, say, height and educational attainment)."

You are still assuming that there is a single dimension being assorted on (even if it is a composite of multiple phenotypes) and not more than one. Seeing as there are strong cross-mate cross-trait correlations across so many phenotypes, e.g. education, height, smoking, alcohol, wealth, psychiatric disorders, etc., it's hard to imagine that these can all be reduced to a single dimension. Who ends up with whom is almost certainly complex.

So despite the increased complexity of the theoretical model, it is still simplistic in some important senses, which is fine, but these limitations need to be acknowledged explicitly. Further, statements about the estimated number of generations of stable assortment need to acknowledge that they rely on these assumptions.

Overall, I believe my issues can be addressed with further revision.

clearly emphasise that the "number of generations of assortment" is only meant as a coarse measure of how close to equilibrium the trait is (i.e., it is *equivalent* to, or *corresponds* to), and is *very* unlikely to reflect the true history of assortment.

In the main manuscript, we have made the following changes (underlined):

Line 236 (p.10) (results-section):

By comparing U_{pgi} to reference values under various heritabilities and assortment strengths, it is possible to infer the equivalent number of generations of stable assortative mating if starting from a random mating population.

Line 336 (p.14) (discussion-section):

In this paper, we estimate that the evidence for educational attainment corresponds to approximately three generations of stable, univariate assortative mating starting from a random mating population, but the exact history of assortment will be longer if the strength of assortment has varied over time or if the genotype-phenotype correlation increased for other reasons⁴⁹.

We have also made some minor changes on page 21 of the Supplementary Information:

First paragraph (before figure):

Panel B in Fig. S8 plots how U develops in successive generations of assortative mating depending on the phenotypic correlation between partners, starting from a random mating population.

Second paragraph (after figure):

By comparing an observed U -value with Fig. S8B, it is possible to infer the equivalent numbers of generations since assortative mating began. Obviously, this would involve making assumptions such as constant environmental variance and constant partner correlations (preceded by random mating), which is unlikely to reflect reality. For this reason, we do not deem it worthwhile to develop a more accurate estimation of t beyond looking up a rough estimate in the figure (although it is perfectly possible to calculate $U(t)$ with other assumptions). A given U can be said to be *equivalent* to an approximate t , which is to say that a population with no assortative mating prior to t generations of constant, univariate assortment would yield a similar U value. It is important to note, though, that multiple other processes could give rise to the same U value, such as changes in assortment strength, changes in environmental and genetic effects across generations, changes in gene-environment correlations, or possibly changing patterns of assortative mating across traits.

Reviewer #2 (Remarks to the Author):

The authors revisions are thorough and the manuscript is much improved. I have no further concerns.